# Pattern recognition in reciprocal space with a magnon-scattering reservoir

Lukas Körber [1,2,5] ✉, Christopher Heins [1,2,5], Tobias Hula[1,3], Joo-Von Kim [4], Sonia Thlang [4], Helmut Schultheiss[1,2], Jürgen Fassbender [1,2] & Katrin Schultheiss[1] ✉

Magnons are elementary excitations in magnetic materials and undergo non-linear multimode scattering processes at large input powers. In experiments and simulations, we show that the interaction between magnon modes of a confined magnetic vortex can be harnessed for pattern recognition. We study the magnetic response to signals comprising sine wave pulses with frequencies corresponding to radial mode excitations. Three-magnon scattering results in the excitation of different azimuthal modes, whose amplitudes depend strongly on the input sequences. We show that recognition rates as high as 99.4% can be attained for four-symbol sequences using the scattered modes, with strong performance maintained with the presence of amplitude noise in the inputs.

A key challenge in modern electronics is to develop low-power solutions for information processing tasks such as pattern recognition on noisy or incomplete data. One promising approach is physical reservoir computing, which exploits the nonlinearity and recurrence of dynamical systems (the reservoir) as a computational resource[1–4]. Examples include a diverse range of materials such as water[5], optoelectronic systems[6–9], silicon photonics[10], microcavity lasers[11], organic electrochemical transistors[12], dynamic memristors[13], nanowire networks[14], and magnetic devices[15–21].

The physical reservoir embodies a recurrent neural network. A natural implementation comprises interconnected nonlinear elements in space (spatial multiplexing, Fig. 1a), where information is fed into the system via input nodes representing distinct spatial elements, and the dynamical state is read out through another set of output nodes[5,12,14,22]. Another approach involves mapping the network onto a set of virtual nodes in time by using delayed-feedback dynamics on a single non-linear node (temporal multiplexing, Fig. 1b)[6–9,13,18], which reduces the complexity in spatial connectivity at the expense of more intricate time-dependent signal processing.

Here, we study an alternative paradigm in which we exploit instead the dynamics in the *modal space* of a magnetic element. This scheme relies on magnon interactions in magnetic materials whereby inputs and outputs correspond to particular eigenmodes of a micromagnetic state. Micrometer-sized magnetic structures can exhibit hundreds of modes in the GHz range[23]. Processes such as three-magnon-scattering interconnect the modes with each other and, with that, provide the nonlinearity and recurrence required for computing. We refer to this approach as modal multiplexing with signals evolving in reciprocal space, in which the actual computation is performed. This is distinct from other wave-based schemes where information is processed with wave propagation and interference in real space[11,16,19,24,25], and differs from temporal multiplexing where virtual nodes are constructed with delayed feedback[6–9,13,18]. The latter also includes reservoirs based on optical cavities where multimode dynamics (such as frequency combs) are exploited but the output spaces are still constructed by temporal multiplexing[26–28].

We illustrate the concept of modal multiplexing with a pattern recognition task using a magnon-scattering reservoir (MSR). The patterns comprise a sequence of symbols "A" and "B" represented by radiofrequency (rf) signals, which consist of sine wave pulses with two distinct frequencies, $f_A$ and $f_B$, and amplitudes $b_{rf,A}$ and $b_{rf,b}$ as shown in Fig. 1d. An example of the power spectrum of the input sequence is

[1]Institut für Ionenstrahlphysik und Materialforschung, Helmholtz-Zentrum Dresden - Rossendorf, Bautzner Landstr. 400, Dresden D-01328, Germany. [2]Fakultät Physik, Technische Universität Dresden, Dresden D-01062, Germany. [3]Institut für Physik, Technische Universität Chemnitz, Chemnitz D-09107, Germany. [4]Centre de Nanosciences et de Nanotechnologies, CNRS, Université Paris-Saclay, 91120 Palaiseau, France. [5]These authors contributed equally: Lukas Körber, Christopher Heins. ✉e-mail: l.koerber@hzdr.de; k.schultheiss@hzdr.de

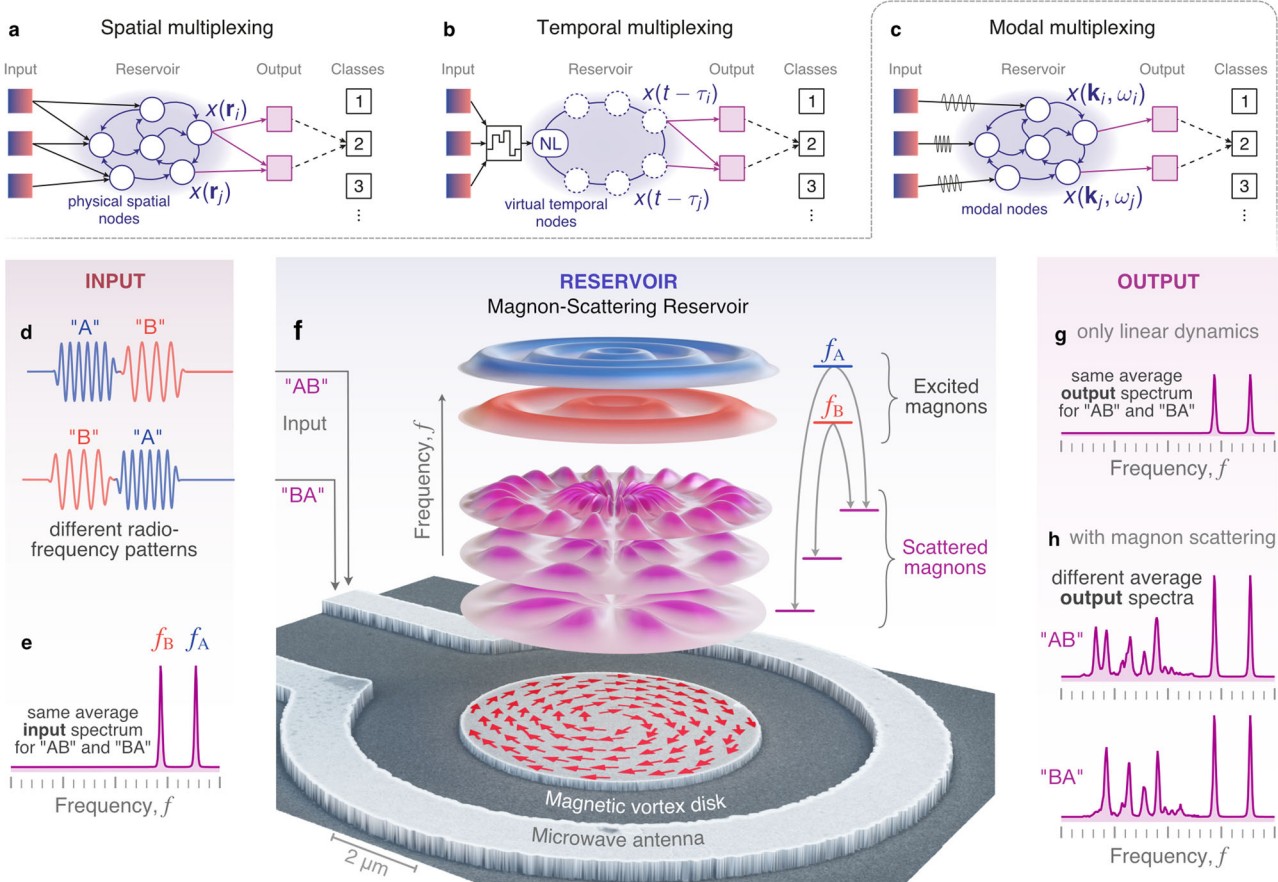

**Fig. 1 | Working principle of a magnon-scattering reservoir (MSR).** Sketches of different reservoirs based on **a** spatial, **b** temporal, and **c** modal multiplexing, the concept behind the MSR. **d** Radiofrequency pulses with different temporal order but **e** the same average frequency content are used to trigger **f** nonlinear scattering between the magnon eigenmodes in a magnetic vortex disk. The dynamic response is experimentally detected using Brillouin-light-scattering microscopy (see Methods). In contrast to a linear system (**g**), the MSR produces different outputs depending on the temporal order of the input (**h**).

given in Fig. 1e. The rf pulses generate oscillating magnetic fields along the $z$ direction through an $\Omega$-shaped antenna, which surrounds a 5.1 $\mu$m wide, 50-nm thick $Ni_{81}Fe_{19}$ disk which hosts a magnetic vortex as a ground state (Fig. 1f). $f_A$ and $f_B$ are chosen to coincide with the frequencies of primary radial eigenmodes of the vortex, which, when excited above a given threshold, result in the excitation of secondary azimuthal eigenmodes through three-magnon-scattering processes[23]. In our previous work[29], we have shown that individual three-magnon splitting channels, e.g. exciting only $f_A$, can be stimulated below their threshold power, and their temporal evolution is significantly modified by additionally exciting one of the secondary modes. In ref. 29, this stimulation was achieved by magnons propagating in a waveguide adjacent to the vortex disk. Here, as a logical extension, the role of the stimulating magnon is provided by the secondary modes of another (active) three-magnon channel $f_B$, a process that we refer to as cross-stimulated three-magnon splitting. The operation of our MSR strongly relies on the fact that the cross-stimulation between $f_B$ and $f_A$ is not reciprocal due to the involved nonlinear transients. In other words, the effect of channel $f_A$ on the channel $f_B$ via cross-stimulation differs from the feedback of $f_B$ on $f_A$.

The power spectrum of excited magnons is obtained experimentally through micro-focused Brillouin light scattering spectroscopy ($\mu$BLS), where a portion of the disk is probed (see Methods and Supplementary Fig. 1a). It is important to note that in the linear response regime neither the input spectrum (Fig. 1e) nor the directly-excited magnon spectrum (Fig. 1g) gives any information about the actual sequence of "A" and "B" (e.g., "AB" and "BA" are equivalent). This

means that no linear classifier can be employed. However, when non-linear processes are at play, magnon-scattering, and associated transient processes result in distinct spectral signatures that can be used to distinguish between different input sequences (Fig. 1h).

## Results

Figure 2 illustrates the role of three-magnon splitting (3MS), the primary nonlinear process at play for the MSR, in which a strongly-excited primary magnon splits into two secondary magnons under the conservation of energy and momentum. In experiments, we choose 20-ns pulses of $f_A$ = 8.9 GHz (20 dBm) and $f_B$ = 7.4 GHz (24 dBm), which excite different radial modes of the vortex above their respective power threshold for 3MS, to represent the symbols "A" and "B", respectively (Fig. 2a). The magnon intensity is probed as a function of frequency and time using time-resolved (TR) $\mu$BLS (see Methods) and is color-coded in Fig. 2b. We measure not only the directly excited primary magnons at $f_A$ and $f_B$, but also magnons at frequencies around half the respective excitation frequencies which result from the nonlinearity of spontaneous 3MS (see Fig. 2c)[23,29]. Here, only the scattering channel with the lowest power threshold is active while other allowed scattering channels remain silent (depicted by dotted lines in Fig. 2c).

Cross-stimulation occurs when signals "A" and "B" overlap in time, as shown in Fig. 2d. Two different primary magnons that share a common secondary mode, as is depicted in Fig. 2f, can result in two 3MS channels that mutually cross-stimulate each other, even below their intrinsic thresholds and along silent channels[29]. Thus, within the overlap interval, the pumped secondary magnon of the first symbol

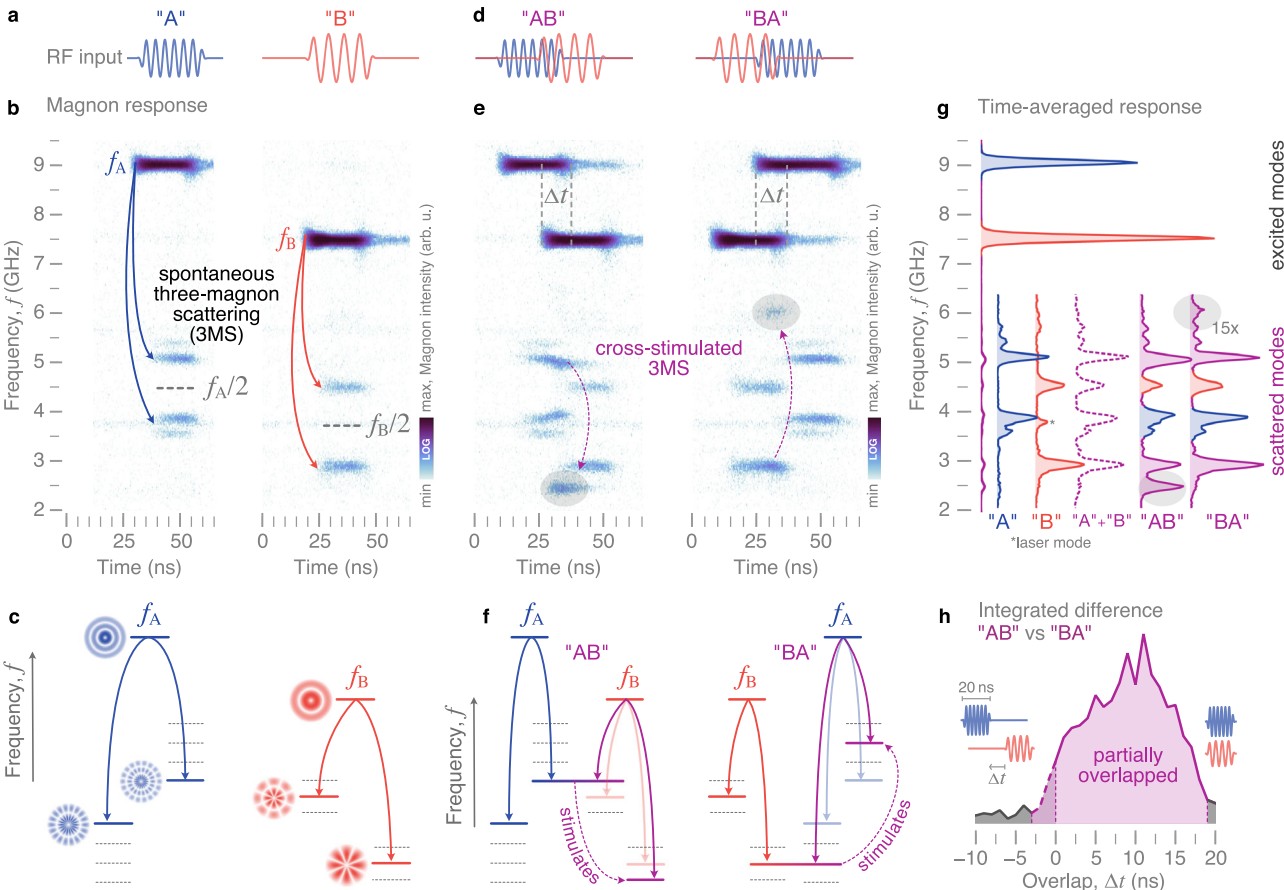

**Fig. 2 | Physical background of the magnon-scattering reservoir (MSR).** When pumped strongly by microwave fields (**a**), a directly-excited primary magnon splits into two secondary magnons (**b**) via spontaneous 3MS (**c**). **b** Time-resolved frequency response of the MSR to two different input frequencies experimentally measured with TR-$\mu$BLS. **d** Driving the MSR with two different, temporally overlapping microwave pulses "A" and "B" leads to **e**, **f** cross-stimulated 3MS between the channels and to additional peaks in the measured frequency response.

**g** Experimentally measured output spectra integrated over time which is different depending on the temporal order of the pulses. Different colors denote different contributions from the two input signals. Blue peaks result from input "A" only, red peaks from "B", and purple peaks from cross-stimulation. **h** The integrated difference between the spectra of "AB" and "BA" shows that the responses are different when the pulses overlap in time.

influences the primary mode scattering of the second symbol, and vice versa, leading to the primary mode scattering into multiple pairs of secondary modes (Fig. 2e).

Because cross-stimulation strongly depends on the temporal order of the primary excitation (Fig. 2f), it provides an important physical resource for processing the temporal sequence of our "AB" signals. This is shown by the experimental results plotted in Fig. 2g, where we compare the time-averaged power spectra for the "AB" and "BA" sequences. These spectra are computed by integrating the temporal data in Fig. 2e. When only signal "A" or only signal "B" is applied, we measure conventional spontaneous 3MS of the respective primary modes with the secondary modes already discussed above in context of Fig. 2b. Within the overlap interval, the mutual cross-stimulation leads to additional peaks in the scattered mode spectrum. As highlighted by shaded areas in Fig. 2e, g, the frequencies and amplitudes of these additional scattered modes strongly depend on the temporal order of the two input signals. Consequently, the average spectra of "AB" and "BA" are different from each other, and neither can be constructed from a simple superposition of the average spectra of "A" and "B" individually (Fig. 2g). This is the key principle that underpins how the MSR processes temporal signals.

To highlight the significance of the transient times, we vary the overlap $\Delta t$ of the symbols "A" and "B" in experiments and determine the frequency-averaged difference between the time-averaged spectra of "AB" and "BA" (Fig. 2h). This difference is zero when the two input

pulses do not overlap since no cross-stimulation takes place. With increasing overlap, however, cross-stimulation between the two pulses becomes more significant and leads to a difference in the output of the reservoir. This difference vanishes again when the input pulses fully overlap and, thus, arrive at the same time.

In order to explore the capabilities of the presented MSR, the complexity of the input signals was further increased experimentally. Figure 3a shows the nonlinear response to the four-symbol pulse pattern "ABAB" measured by TR-$\mu$BLS. In contrast to a reference spectrum composed by a simple linear superposition of two consecutive "AB" patterns, shown in Fig. 3b, the real spectrum of the four-symbol response contains additional features which are generated by cross-stimulated scattering when two pulses overlap. The differences are highlighted by the shaded areas in Fig. 3a and circled areas in Fig. 3b, respectively. This behavior illustrates that cross-stimulation can result in distinct features that allow distinguishing also longer patterns. This is further exemplified in Fig. 3c, which shows the time-averaged BLS spectra of the six four-symbol combinations comprising two "A" and two "B". Like the data in Fig. 2, transient processes from cross-stimulation generate distinct power spectra for the six combinations, which would be indistinguishable in the linear response regime.

Since the experimental data discussed so far requires the integration of thousands of pulse cycles, we rely on micromagnetic simulations to quantify the capacity of the MSR for recognizing all

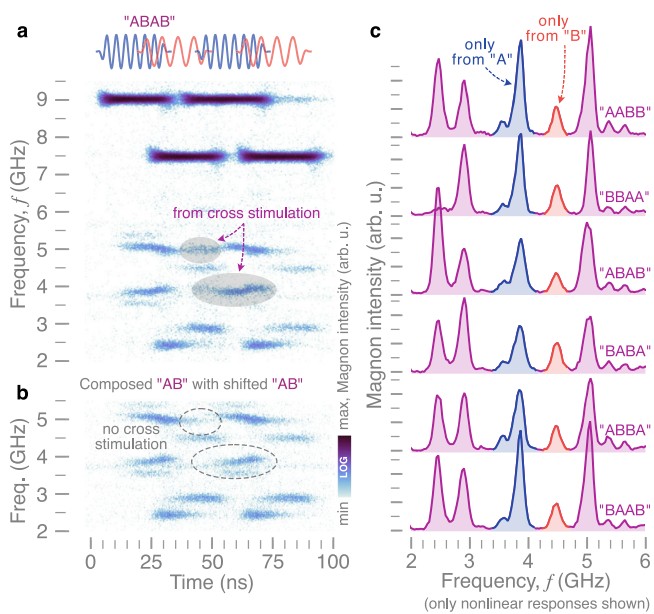

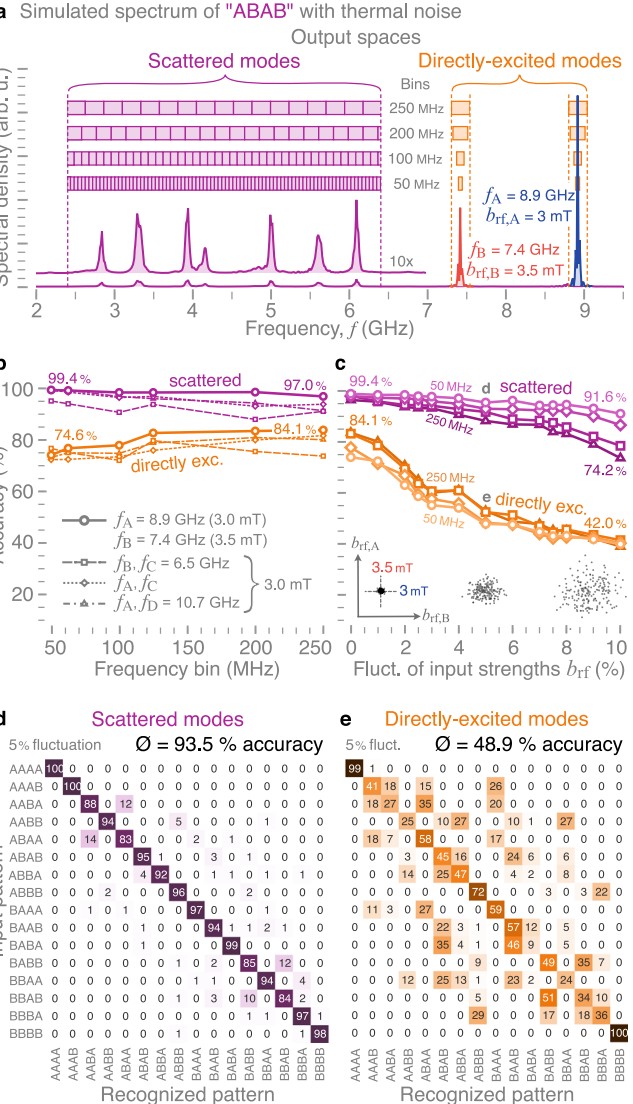

**Fig. 3 | Performance of the MSR for longer temporal patterns characterized experimentally. a** Time-resolved spectral response of the MSR to a four-symbol microwave pattern "ABAB", detected experimentally with TR-$\mu$BLS. **b** For reference, the spectrum of "AB" is overlayed with a shifted version of itself. Differences between composed and real spectrum (due to cross-stimulated magnon scattering) are highlighted by shaded and circled areas. **c** Average output spectra of the MSR for different four-symbol patterns with the same average input-frequency content but clearly different nonlinear responses.

**Fig. 4 | Micromagnetic modeling of pattern recognition capabilities.**
**a** Simulated spectrum of the pattern "ABAB" with the definition of different output spaces (scattered and directly excited modes) for the MSR. **b** Average detection accuracy of four-symbol patterns for different output spaces and excitation frequency and power combinations as a function of frequency bin sizes. **c** Accuracy for different output spaces and bin sizes as a function of power fluctuations in the input signals (depicted by the insets). **d**, **e** Corresponding confusion matrices for the two output spaces, respectively, both for the same frequency combination, bin size, and input power fluctuation.

possible combinations of four-symbol sequences composed from "A" and "B" (see Methods). Thereby, we are able to analyze individual pulse sequences and study the influence of thermal noise and amplitude fluctuations on the recognition rate of the MSR. Figure 4a shows a simulated power spectrum (at $T = 300$ K) for the input pattern "ABAB" with $f_A = 8.9$ GHz ($b_{rf,A} = 3$ mT) and $f_B = 7.4$ GHz ($b_{rf,B} = 3.5$ mT), with the field strengths chosen to be above the respective power threshold for 3MS. The output spaces of the reservoir are defined by subdividing the time-averaged power spectrum into frequency bins of different widths. To emphasize the importance of the scattering (interconnection) between the different magnon modes, we study the performance of the MSR for two separate output spaces (Fig. 4a). One output space for the scattered modes is constructed over a 4-GHz window below $f_A$ and $f_B$, where the different frequency bins result in an output vector with 16–80 components depending on the bin size (see Supplementary Note 1). For comparison, a two-dimensional output space corresponding to the directly-excited modes is constructed by averaging within bins centered around $f_A$ and $f_B$. Note that, here, analyzing the directly-excited modes does not correspond to a linear classifier, as these modes themselves experience nonlinear feedback (amplitude losses, frequency shift, etc.) above their power threshold for 3MS.

For each four-symbol sequence, 200 micromagnetic simulations were executed with different realizations of the thermal field in order to generate distinct output states. Supervised learning using logistic regression was then performed on this data set to construct trained models of the output states based on either the directly-excited or scattered modes. The accuracy of these models for different combinations of input frequencies $f_A = 8.9$ GHz, $f_B = 7.2$ GHz, $f_C = 6.5$ GHz, $f_D = 10.7$ GHz (and corresponding input strengths $b_{rf,i}$) is shown in Fig. 4b as a function of bin size. We find that the MSR performs comparably well when choosing different input frequencies (different radial modes) to represent the input symbols. Hence, an

extension of the input space to more than two frequencies/symbols ("ABC", "BDC", "ABCD", etc.), or even to more broadband signals, is straightforward. To this end, Supplementary Note 2 contains the measured and simulated distinct nonlinear responses for different permutations of the three and four-frequency sequences "ABC" and "ABCD".

Overall, the accuracy depends weakly on the bin size. The recognition rate slightly increases with increasing bin size for the directly-exited modes whereas it decreases marginally for the scattered modes. This can be understood from the fact that smaller bin sizes capture more features of the power spectrum of the scattered modes, while for the directly-excited modes, the larger bin sizes contain more information about potential nonlinear frequency shifts, which helps to separate the inputs. We observe that outputs based on the directly-excited modes can yield an accuracy of ~84%, while

scattered modes provide a notable improvement in performance, with an accuracy reaching 99.4% for the case considered in Fig. 4a.

In general, the scattered modes provide higher accuracy for pattern recognition compared with the directly-excited modes. The difference in accuracy becomes even more pronounced when amplitude fluctuations are present. Figure 4c illustrates how the accuracy evolves with the fluctuation strength, which represents the width of the normal distributions (in %), centered around the nominal values of $b_{\text{rf,A}}$ and $b_{\text{rf,B}}$, from which the field strengths are drawn, as shown in the inset for $b_{\text{rf,A}} = 3$ mT and $b_{\text{rf,B}} = 3.5$ mT. The performance of the MSR based on the directly-excited modes drops significantly with increasing fluctuation strength (42% accuracy at 10% fluctuation). However, recognition based on the scattered modes is much more resilient, with a decrease to only between ~74% and ~92% accuracy (depending on the bin size).

Figure 4d, e show confusion matrices for the scattered and directly-excited modes, respectively, both for the same set of parameters. They highlight the robustness of the MSR which is based on the scattered modes since it mainly fails to distinguish "AABA" from "ABAA" and "BBAB" from "BBAB" in ~12% of the cases. The MSR based on the directly-excited modes, on the other hand, fails to recognize almost all of the patterns, except for the trivial cases of "AAAA" and "BBBB" for which there is practically no ambiguity in the inputs. These trends do not depend on the type of supervised learning used and highlight the important role of cross-stimulated 3MS in the MSR for the pattern recognition of noisy radiofrequency signals.

## Discussion

Our findings demonstrate the possibility of performing reservoir computing in modal space utilizing the intrinsic nonlinear properties of a magnetic system, namely the scattering processes between magnons in a magnetic vortex disk. Temporal patterns encoded using two different input-frequency pulses can be distinguished with high accuracy. The results also indicate that input patterns can be extended to more broadband signals. We note that the technical design of the physical reservoir is extremely simple and requires very little pre-possessing, while the complexity of the data handling arises mostly from the intrinsic nonlinear dynamics of the magnon system. Additionally, recent findings have shown that the magnon interactions in micrometer-sized disks can be modified significantly by small static magnetic fields[30], providing effective means to enhance the complexity of the magnon-scattering reservoir further. Although our current read-out scheme is based on optical methods, magnetoresistive sensors hold promising possibilities for an all-electric read-out.

## Methods

### Sample preparation and characterization

The magnetic disk housing the magnon-scattering reservoir for our experiments was manufactured in a two-step procedure: In a first step, using electron-beam evaporation and subsequent lift off, a magnetic disk with a diameter of 5.1 μm was patterned from a $Ti(2)/Ni_{81}Fe_{19}(50)/Ti(5)$ film deposited on a $SiO_2$ substrate which had been capped with a 5-nm thick aluminum layer. All thicknesses are given in nanometers. In a second step, an Ω-shaped antenna used to excite magnon dynamics in the reservoir was patterned around the disk from a Ti(2)/Au(200), also using electron-beam evaporation and subsequent lift off. The inner and outer diameter of the antenna are 8.3 μm and 11.1 μm, respectively. An image of the sample, obtained with scanning electron microscopy, can be seen in Supplementary Fig. 1a.

### Signal generation

The radiofrequency (rf) pulses were generated by two separate rf-sources set to a fixed frequency corresponding to pulse "A" and pulse "B", respectively (see Supplementary Fig. 1b). In order to synchronize the two sources, a pattern generator (Pulsestreamer by Swabian Instruments) was used to create a pattern of arbitrary shape gating the

rf-sources. The two generated microwave signals were combined and fed onto the Ω-shaped antenna using picoprobes.

### Time-resolved Brillouin-light-scattering microscopy

All experimental measurements were carried out at room temperature. Magnon spectra were obtained by means of Brillouin-light-scattering microscopy as schematically shown in Supplementary Fig. 1b[31]. A monochromatic 532-nm laser (CW) was focused onto the sample surface using a 100x microscope lens with a numerical aperture of 0.7. The backscattered light was then directed into a Tandem Fabry-Pérot interferometer (TFPI) using a beam splitter (BS) in order to measure the frequency shift caused by inelastic scattering of photons and magnons. Control signals that encode the current state of the interferometer, signals of the photon counter inside the TFPI and a clock signal from the pattern generator were acquired continuously by a time-to-digital converter (Timetagger 20 by Swabian Instruments). From these data, the temporal evolution of the magnon spectra with respect to the stroboscopic rf excitation was reconstructed. During the experiments, the investigated structure was imaged using a red LED and a CCD camera (red beam path in Supplementary Fig. 1b). Displacements and drifts of the sample were tracked by an image recognition algorithm and compensated by the positioning system (XMS linear stages by Newport). The laser and imaging beam path were separated by the dichroic mirror as shown in Supplementary Fig. 1b. In order to ensure that all stationary magnon modes were measured, the signal was averaged over 10 positions across half the disk as seen in Supplementary Fig. 1a.

### Micromagnetic simulations

Simulations of the vortex dynamics were performed using the open-source finite-difference micromagnetics code MuMax3[32], which performs a time integration of the Landau-Lifshitz-Gilbert equation of motion of the magnetization $\mathbf{m}(\mathbf{r}, t)$,

$$\frac{\partial \mathbf{m}}{\partial t} = -\gamma \mathbf{m} \times (\mathbf{B}_{\text{eff}} + \mathbf{b}_{\text{th}}) + \alpha \mathbf{m} \times \frac{\partial \mathbf{m}}{\partial t}. \quad (1)$$

Here, $\mathbf{m}(\mathbf{r}, t) = \mathbf{M}(\mathbf{r}, t)/M_s$ is a unit vector representing the orientation of the magnetization field $\mathbf{M}(\mathbf{r}, t)$ with $M_s$ being the saturation magnetization, $\gamma = g\mu_B/\hbar$ is the gyromagnetic constant, and $\alpha$ is the dimensionless Gilbert-damping constant. The effective field, $\mathbf{B}_{\text{eff}} = -\delta U/\delta \mathbf{M}$, represents a variational derivative of the total magnetic energy $U$ with respect to the magnetization, where $U$ contains contributions from the Zeeman, nearest-neighbor Heisenberg exchange, and dipole-dipole interactions. The term $\mathbf{b}_{\text{th}}$ represents a stochastic field with zero mean, $\langle b_{\text{th}}^i(\mathbf{r}, t) \rangle = 0$ and spectral properties satisfying[33]

$$\left\langle b_{\text{th}}^i(\mathbf{r}, t) b_{\text{th}}^j(\mathbf{r}', t') \right\rangle = \frac{2\alpha k_B T}{\gamma M_s V} \delta_{ij} \times \delta(\mathbf{r} - \mathbf{r}')\delta(t - t'), \quad (2)$$

with amplitudes drawn from a Gaussian distribution. Here, $k_B$ is Boltzmann's constant, $T$ is the temperature, and $V$ denotes the volume of the finite-difference cell. This stochastic term models the effect of thermal fluctuations acting on the magnetization dynamics. An adaptive time-step algorithm based on a sixth-order Runge-Kutta-Fehlberg method was used to perform the time integration[34].

We model our 50-nm thick, 5.1-μm diameter disk using $512 \times 512 \times 1$ finite-difference cells with $\gamma/2\pi = 29.6$ GHz/T, $M_s = 810$ kA/m, an exchange constant of $A_{\text{ex}} = 13$ pJ/m, and $\alpha = 0.008$. Previous work has shown that these simulation parameters provide excellent agreement with previous experimental results[23].

For the magnon dynamics shown in Fig. 4, we first obtain the magnetic ground state of the disk by initializing with a vortex state and subsequently relaxing the magnetization by minimizing the total magnetic energy in the absence of any static external applied fields.

Magnons are then excited under a finite temperature of 300 K. First, we let the system evolve for 5 ns under the action of thermal fluctuations alone. A spatially uniform oscillating magnetic field $\mathbf{b}_{\mathrm{rf}}(t) = b_{\mathrm{rf}}(t)\mathbf{e}_z$ is then applied along the $z$ direction, perpendicular to the film plane. Following the experimental work, the 4-symbol pulse patterns are encoded into $\mathbf{b}_{\mathrm{rf}}(t)$ as a combination of two input-frequency signals,

$$b_{\mathrm{rf}}(t) = W_A(t)b_{\mathrm{rf},A}\sin(2\pi f_A t) + W_B(t)b_{\mathrm{rf},B}\sin(2\pi f_B t). \quad (3)$$

$W_A(t)$ and $W_B(t)$ represent windowing functions where each "A" or "B" pattern lasts 20-ns with a 5-ns overlap between patterns. These windowing functions are illustrated in Supplementary Fig. 2 for the 16 4-symbol pulse patterns considered. The excitation field amplitude $b_{\mathrm{rf},i}$ and frequency $f_i$ for each pattern is given in the main text. After the end of the last pattern, the transient dynamics is computed for an additional 10 ns. The dynamics is simulated for a total duration of 80 ns for each 4-symbol pattern.

The power spectral density of the magnon excitations is computed by using a coarse-graining procedure (Supplementary Fig. 3).

The simulation geometry is further sub-divided using a triangle mesh (Supplementary Fig. 3a) in the film plane whereby we record the spatial average of the magnetization vector as a function of time, i.e., for a mesh element $j$,

$$\mathbf{m}_j(t) = \frac{1}{V_j}\int_{V_j} d^3x\, \mathbf{m}(\mathbf{r},t), \quad (4)$$

where $V_j$ is the volume of the mesh element. With $V = \sum_j V_j$ representing the total volume, the total power spectral density $\mathcal{S}(\omega) = (1/V)\sum_j \mathcal{S}_j(\omega)V_j$ is then constructed from the discrete Fourier transform of the $z$ component of the averaged magnetization for each element, $\mathcal{S}_j(\omega) = |\mathcal{M}_j(\omega)|^2$, where

$$\mathcal{M}_j(\omega) = \sum_{n=0}^{N-1} e^{-i\omega(n\Delta t)}m_{j,z}(n\Delta t), \quad (5)$$

$\Delta t = 20$ ps is the time-step, and $N = 8000$ is the total number of time steps. Supplementary Fig. 3a and b illustrate the power spectrum for individual regions, i.e., for Region 1 and Region 2 in Supplementary Fig. 3a, respectively. Even at the level of a single mesh region, we can clearly identify the directly excited modes at $f_A$ and $f_B$, along with a number of scattered modes. Averaging over a quadrant of the disk gives the power spectrum in Supplementary Fig. 3d, where we can see a much-improved signal-to-noise ratio of the excited and scattered modes. Supplementary Fig. 3e shows the power spectrum averaged over all the mesh regions of the disk, which is qualitatively very similar to the quadrant-averaged result in Supplementary Fig. 3d. For this reason, we only used the quadrant-averaged spectra for the pattern recognition tasks in the interest of minimizing computation time without loss of generality. The construction of the output spaces from the obtained spectra is described in Supplementary Note 1.

## Data availability
The numerical and experimental data used in this study are available in the RODARE database under https://doi.org/10.14278/rodare.2344.

## Code availability
The software package used for micromagnetic simulations is found at http://mumax.github.io/api.html.

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

## Acknowledgements

The authors are thankful to D. Rontani and K. Knobloch for providing feedback on the manuscript and fruitful discussions. This study was supported by the German Research Foundation (DFG) within programs SCHU 2922/1-1 (H.S., T.H., C.H.), KA 5069/1-1, and KA 5069/3-1 (L.K.), as well as by the French Research Agency (ANR) under contract No. ANR-20-CE24-0012 (MARIN) (S.T., J.V.K.). The project has received funding from the EU Research and Innovation Programme Horizon Europe under grant agreement no. 101070290 (NIMFEIA) (K.S.). Support by the Nanofabrication Facilities Rossendorf (NanoFaRo) at the IBC is gratefully acknowledged.

## Author contributions

Conceptualization: H.S., J.V.K., K.S. Investigation: C.H. Simulation: J.V.K., S.T. Visualization: L.K., T.H., J.V.K., H.S. Funding acquisition: J.V.K., H.S., J.F. Project administration: H.S., K.S. Writing–original draft: L.K. Writing—review and editing: L.K., C.H., T.H., J.V.K., H.S., J.F., K.S.

## Funding

## Competing interests

The authors declare no competing interests.
