## [Peer Review File · Nature Communications]

Reviewers' Comments:

Reviewer #1:

Remarks to the Author:

The paper of Körber et al is a beautiful demonstration of a computing concept where the dynamics of coupled nonlinear magnetic eigenmodes performs neuromorphic computing. The paper experimentally demonstrates that the coupled mode dynamics acts as a reservoir and significantly enhances the recognition capability of a neural network when inserted before a linear reservoir.

What I find the most appealing about the paper:

It is experimental demonstration and, in a sense, an experimental tour de force. The authors have a clean, deep understanding of the magnetization dynamics of the system they study.

The paper is a first in demonstrating a neural network in the modal space, which is a potentially breakthrough concept. Thermal noise is also discussed, which is a big plus.

What I find problematic in the paper:

It is not at all clear that this system is useful practically or that it may eventually lead to useful systems. Magnetic systems have an expensive I/O overhead: this simple magnetic reservoir will become an extremely complex device if you will access the modes electrically (as opposed to the optical / BLS way it is done now). This still would be ok – but I just do not see a way of scaling up the system to a size that would amortize the I/O overhead. A magnetic disk will only have a very limited number of modes (I would say no more than ten or so), which will translate to a very little number of effective neurons.

Also, due to experimental challenges, some results are obtained by simulations. How this will become a useful chip-scale device if it is difficult to study even by expensive lab tools?

The authors should, at the minimum compare or benchmark this device to an electrical or optical reservoir and at a minimum they show that a scaled-up system is physically possible and would be attractive in terms of performance. What about saying something about the number of internal (nonlinear) interconnections / neurons, characterizing the nonlinearity, understanding the role of excitation intensity in the device dynamics? And do this also with the assumption that many more neurons (i.e. reservoir complexity) would be needed.

There are a few ambiguities in the paper that should be more explicitly explained.

Most importantly some results are obtained from simulations others are from experiments. I myself could not always follow what comes from where. Would it be possible to devote a short paragraph to clarify this, and/or carefully state in each figure caption what comes from simulations and what does not?

The abstract claims 95% accuracy while the text claims 99.4% at the end. Please clarify what comes from which model.

Also, reservoir computing devices are notably tricky to characterize: even a very poorly-performing reservoir may seem to achieve high accuracy because of the output linear classifier alone works rather well for many problem classes. I understand that the authors check the role of scattered vs directly accessed nodes to understand this problem – perhaps this argument should be a bit more explicit.

Overall, I very much enjoyed reading the paper and the scientific quality of the work would deserve Nat. Comm. publication. I have more reservations about the practicality of the device and what future and impact it would have in future computing devices. At a minimum these should be addressed by a few additional arguments in the paper.

Reviewer #2:

Remarks to the Author:

This paper studies physical reservoir computing utilizing temporal response of a spin vortex to a time-series input signal, where their unique approach is to use time-averaged power spectra of spin motions for machine-learning computing to extract information encoded in input signals. The

former part describes experimental results to demonstrate the basic operation principal of their computing scheme, whereas the latter part describes micromagnetic simulation results and reservoir computing demonstrations.

In the first experimental part, the authors prepared a permalloy disk with an outer ring electrode to induce a time-series rf magnetic field as input. Spin motions at 10 different positions were detected by micro-focused Brillouin light scattering and they were averaged to obtain a power spectrum for output. An input is encoded using a specific combination of the amplitude and frequency of a 20-ns-long sinusoidal signal and two inputs "A" and "B" were prepared. From the end time of an input, a time-series spin motion was recorded for 10 ns to obtain a time-averaged power spectrum. Owing to spontaneous three-magnon scattering and characteristic modes of spin vortex, "A" and "B" have different characteristic spectra having a component at the input frequency as well as two components at lower frequencies. Next, "A" and "B" ("B" and "A") are sequentially combined with an overlap time, to prepare "AB" ("BA") input. Interestingly, an additional component that does not appear in "A" or "B" was clearly obtained. Thus, "AB" and "BA" can be distinguished by power spectra. This is the key to their computing.

In the next part, spin motions in the same permalloy disk were simulated using micromagnetics, with taking into account thermal noise at 300 K and input signal fluctuations. In addition to the experimental part, two inputs "C" and "D" were also prepared, and four-series inputs (ex. ABAB) were explored. A characteristic power spectrum was obtained for each input combination. Using power spectra for all the combinations, reservoir computing was performed to classify series inputs. In consequence, a high classification performance is achieved even under moderate noise levels and input fluctuations.

Their original idea to use time-averaged spectra for reservoir computing has a high impact, their experiments are fundamentally solid, and the high classification accuracy is interesting and remarkable. On the other hand, I think that there are some unclear descriptions, lack of explanations for procedures, and speculative outlooks. Hence, I basically recommend for publication if the manuscript is appropriately/satisfactorily revised.

(1)

To show the overall of the "reservoir computing" research field, Ref. [1] is just described. I think this reference paper is basically liquid state machine developed from neuroscience and a pulse-based computing model. In addition, the word "reservoir computing" covers analog computing models, such as echo state networks, developed from artificial recurrent neural networks. In my opinion, the authors should show "Jaeger, GMD technical report 148, 2001.". This also comes from the fact that the authors used analog computing, not pulse-based computing.

Physical reservoir computing was derived from reservoir computing, however, the relation between those two computing frameworks is yet unclear. In my opinion, showing references on physical reservoir computing is very helpful for readers, if possible. For example, G. Tanaka et al., Neural Networks 2019, K. Nakajima, Jpn. J. Appl. Phys. 2020.

If the authors should choose one due to the length limit, I think Jaeger is appropriate according to the computational method in this paper.

(2)

I understand that cross-stimulated three-magnon scattering (3MS) is the key for successful computing. For example, in Fig. 2E, the frequencies of 3MS for "AB" and "BA" are different from each other. What is the physics behind those phenomena? What are the roles of the first and second inputs? In addition, this type of temporal 3MS has been reported so far? In my opinion, the authors should give comments on these, which can also maximize the originality of this work even the explanation of the physics is qualitative (or speculative). I know that analyses on temporal dynamics are usually difficult, but the knowledge can contribute to understanding of physical reservoir computing.

(3)

Throughout the paper, the authors do not describe the computing system in terms of reservoir computing framework. A reservoir computing consists of a preprocessing part, a reservoir part,

and a readout part. The input encoding corresponds to the preprocessing part, and the disk corresponds to the reservoir part. These are easy to understand despite of explicit explanations. However, "a power spectrum" corresponds to post signal processing that is contained in the readout part. Also, the worst thing is that the authors do not explain how to obtain the system output class, particularly, there is no description about the readout part. In general sense, there is an output layer with the same size as the target combination numbers, all the reservoir outputs are connected to each system output node, and there are connection weights for all the connections. In S.M., the authors just describe "we used the classify function", but readers cannot understand the entire system design. Also, to classify some labels, there are many methods to use, for example, setting one hot vector, the maximum output level in the testing phase is selected, and finally its class label is used for the prediction class. In the present style, readers including me do not understand what are done in the machine learning computing. Hence, I think the authors should describe a simple explanation in the main text and details in S.M..

(4)

In Fig. 3C, there are many spectra having different shapes. I think it is very helpful for readers to assign the feature induced by 2-series inputs on each typical peak, for example, "AB" and "BA". This is because the computing seemingly captures the spectrum characterized by the peak amplitude differences and readers must want to simply know what are different.

(5)

In Fig. 4A, the spectrum for "ABAB" is shown. However, I think this is different from the spectrum in Fig. 3C. Please check this. Also, why the frequency range in Fig. 3C is different from that in Fig. 4A? The same range should be appropriate.

(6)

In page 5, in the right column, in the first paragraph

"Hence, an extension of the input space to more than two frequencies/symbols, or even to more broadband signals, is straightforward."

This is unclear to me. If the authors claim that they can use 3 or more input expressions for an input sequence (ex. "ABCD"), I do not agree with this. If the authors claim that they can use large numbers of inputs in an input sequence (ex. "ABABABAB"), I do not agree with this.

For the former, this is because the computing basically uses temporal nonlinear phenomena and thus one cannot tell what happens when unknown series inputs are used for an input sequence. In the manuscript, the authors just use two inputs for input sequences, which corresponds to binary bit computing. Hence, the authors did not clearly show that their computing method works well for fully analog computing, and thus the outlook is too speculative.

For the latter, this is because a power spectrum is calculated using a 10-ns-long transient signal. If there are many sequences, some components will be overlap and/or long-past memories have relatively smaller components. The authors demonstrated only four inputs sequences. Hence, the authors did not clearly show that their computing method works well for long-range input sequences and thus the outlook is too speculative.

I would like the authors to describe more clearly what they want to say.

(7)

Throughout the manuscript, the authors do not describe important features for reservoir computing: nonlinearity, short-term memory, and variability. Nonlinearity do not mean "nonlinear phenomena". It is defined by nonlinear temporal transformation of input vector to reservoir output vector. This is usually proven by nonlinear tasks, such as temporal XOR. Variability is more difficult to understand for physical reservoir computing researchers, so usually it is not discussed. I think those two are difficult to discuss in the present reservoir computing system and input sequences, and thus it is OK to neglect them. However, I think short-term memory plays an important role in the present system. For example, "ABBA" has a different spectral feature from "AABA", meaning that first "AB" and "AA" are memorized in their spectra, so I think the memory at least three-step past is contained, i.e., memory length more than 3. Like this, please discuss the results in the context of reservoir computing. (Actually, if one cannot show success in nonlinear tasks, it cannot

be called true "reservoir computing".)

Reviewer #3:

Remarks to the Author:

L. Korber et.al. reported a pattern recognition method using a magnon reservoir. By using a nonlinear 3-magnon scattering, the authors realized recurrent system in single magnetic disk. The modal multiplexing was clearly detected by using time resolved micro-BLS measurement. The recognition accuracy up to 95% was presented. The wave-based reservoir is important research field of magnonics, and this result is interesting. The contents of this paper are well organized and understandable. The reviewer inquires information to judge its experimental validity. The points are listed below.

1. For the modal multiplexer, the probe-position dependence of recognition accuracy appears to be of small importance. However, if the probing position was changed, what occurs for the recognition accuracy? If the modal or magnon states was affected by nonlinear 3-magnon scattering, the wave patterns could be different for probing positions and thus the BLS spectrum also could be changed. The presented data were probed by symmetrical positions given in SM file; however, the antisymmetric position appears to increase the accuracy. On the contrary, if the symmetrical position was better, the wave patterns appear to be a steady and less affected by the nonlinear effect (thus history). Does this mean that the system works as physical reservoir with small history?

2. For the physical reservoir, the output is also important. In Fig. 1F, the output node is not shown. For the actual system, the laser-based output or detection method appears to be difficult (BLS or Kerr equipment for single disk). The importance of this paper is the observation of multiplexer in the reciprocal space. However, if the authors insist this method is simple and requires very little preprocessing, the authors discuss the complexity of detection (output) node for modal multiplexing.

3. In Fig. 4B, some description is lacked for f_A , f_C (no frequency information was given).

4. The authors considered the amplitude noise about 3.0mT. Whys this value was picked up? The amplitude dependence of accuracy should be discussed.

Reviewer #1

The Reviewer writes:

The paper of Körber et al is a beautiful demonstration of a computing concept where the dynamics of coupled nonlinear magnetic eigenmodes performs neuromorphic computing. The paper experimentally demonstrates that the coupled mode dynamics acts as a reservoir and significantly enhances the recognition capability of a neural network when inserted before a linear reservoir.

What I find the most appealing about the paper:

It is experimental demonstration and, in a sense, an experimental tour de force. The authors have a clean, deep understanding of the magnetization dynamics of the system they study.

The paper is a first in demonstrating a neural network in the modal space, which is a potentially breakthrough concept. Thermal noise is also discussed, which is a big plus.

We thank the reviewer for their positive appraisal of our work.

What I find problematic in the paper:

It is not at all clear that this system is useful practically or that it may eventually lead to useful systems. Magnetic systems have an expensive I/O overhead: this simple magnetic reservoir will become an extremely complex device if you will access the modes electrically (as opposed to the optical / BLS way it is done now). This still would be ok – but I just do not see a way of scaling up the system to a size that would amortize the I/O overhead. A magnetic disk will only have a very limited number of modes (I would say no more than ten or so), which will translate to a very little number of effective neurons.

The Reviewer raises some interesting points about integration into conventional CMOS electronics, which would be applicable to any potential future technology based on magnetism, spintronics, or any other exotic materials that do not feature in any current industrial process. However, we believe it is too early to speculate on what form a magnon-scattering reservoir would like in an integrated circuit. Our goal here is to discuss a *new paradigm* in reservoir computing type approaches to artificial neural networks, where the nodes are represented by spin-wave modes and massive connectivity is established in *reciprocal space*, rather than in real space or in the time domain. This paper represents the starting point in the exploration of such a concept, so we are confident that issues related to addressing the modes electrically, for example, will be tackled as more researchers turn their attention toward this problem.

The Reviewer's assertion on the limited number of modes available ("ten or so") is not quite correct. Micrometer-sized disks exhibit *hundreds* of modes in the GHz range, a few of which are plotted in Fig. R1(c) below [see also Fig. 1(b) in K. Schultheiss et al., *Phys. Rev. Lett.* **122**, 097202 (2019)]. By exciting radial modes with slightly different frequencies, different 3-magnon splitting channels can be addressed, as is shown in [R. Verba et al., *Phys. Rev. B* **103**, 014413 (2021)]. Overall, this leads to complex nonlinear dynamics within one single disk already.

Helmholtz-Zentrum
Dresden–Rossendorf e. V.

Address:
Bautzner Landstr. 400
D-01328 Dresden
<http://www.hzdr.de>

Board of Directors:
Prof. Dr. Sebastian M. Schmidt
Dr. Diana Stiller

Company Registration Number:
VR 1693, Amtsgericht Dresden

Bank Details:
Commerzbank AG
Account No. 0402 657 300
(Bank Code 850 800 00)
BIC DRESDEFF850
IBAN DE42 8508 0000 0402 6573 00

VAT-ID-No.: DE140213784

Fig. R1: Magnon frequencies as a function of radial mode number n and azimuthal mode number m obtained from micromagnetic simulations for NiFe disks with diameters of (a) 1 μm , (b) 2 μm , and (c) 5.1 μm . Each data point resembles one magnon mode. Taken from [L. Körber, Master Thesis, Technische Universität Dresden (2019)]

Furthermore, it is straightforward to alter the excitation scheme and pattern a multitude of disks on the same strip-line antenna. Disks with varying diameters [Figs. R1(a), R1(b)], thicknesses, and even magnetic materials exhibit vastly different magnon modes and 3-magnon scattering channels. Even different geometries can be envisioned. Considering all the different magnetic elements as part of one reservoir enhances its complexity even more.

Additionally, we recently demonstrated in [L. Körber et al., *Appl. Phys. Lett.* **122**, 092401 (2023)] that the application of in-plane magnetic fields in the order of a few mT modifies 3-magnon splitting further. When patterning disks on a strip-line antenna, such field amplitudes can be easily achieved by the application of direct current pulses. This can serve as an additional means to increase the complexity of the magnon-scattering reservoir.

Regarding the practical aspects, these will be pursued within the framework of a recently-funded project by the European Commission (<https://cordis.europa.eu/project/id/101070290>). Part of the aims of this project is to achieve electrical read-out using magnetoresistive sensors compatible with industrial processes. The fact that two major industrial partners are on-board with this effort illustrates the strong potential for real applications. However, such discussions are beyond the scope of this manuscript.

To highlight the importance of the number of modes and the practical read out, we added the following sentences to the introduction of our manuscript:

Micrometer-sized magnetic structures can exhibit hundreds of modes in the GHz range [*Phys. Rev. Lett.* **122**, 097202 (2019)].

Helmholtz-Zentrum
Dresden-Rossendorf e. V.

Address:
Bautzner Landstr. 400
D-01328 Dresden
<http://www.hzdr.de>

Board of Directors:
Prof. Dr. Sebastian M. Schmidt
Dr. Diana Stiller

Company Registration Number:
VR 1693, Amtsgericht Dresden

Bank Details:
Commerzbank AG
Account No. 0402 657 300
(Bank Code 850 800 00)
BIC DRESDEFF850
IBAN DE42 8508 0000 0402 6573 00

VAT-ID-No.: DE140213784

and the following sentence to the conclusion:

Additionally, recent findings have shown that the magnon interactions in micrometer-sized disks can be modified significantly by small static magnetic fields [L. Körber et al., *Appl. Phys. Lett.* **122**, 092401 (2023)], providing effective means to enhance the complexity of the magnon-scattering reservoir further. Although our current read-out scheme is based on optical methods, magnetoresistive sensors hold promising possibilities for an all-electric read-out.

Also, due to experimental challenges, some results are obtained by simulations. How this will become a useful chip-scale device if it is difficult to study even by expensive lab tools?

Again, this manuscript provides the first demonstration of a new concept, so we would argue that it is unreasonable to expect that all technological issues have also been solved. If we draw upon developments in magnetism and spintronics over the past 20 years as a guide, we note that similar criticisms could equally have been leveled at spin-torque-induced switching, which now forms the basis of commercially-available spin-transfer-torque random access memories (STT-RAM), and spin-torque induced oscillations, which form the basis of spin-torque nano-oscillators (used in microwave-assisted magnetic recording technology, for example). Initial reports on these phenomena were also conceptual in nature, obtained using expensive microwave characterization tools, and presented the same kinds of (perceived) difficulties as the Reviewer mentioned. In this light, publication in *Nature Communications* would help disseminate our concept to a broad audience and stimulate further work on the topic.

The authors should, at the minimum compare or benchmark this device to an electrical or optical reservoir and at a minimum they show that a scaled-up system is physically possible and would be attractive in terms of performance. What about saying something about the number of internal (nonlinear) interconnections / neurons, characterizing the nonlinearity, understanding the role of excitation intensity in the device dynamics? And do this also with the assumption that many more neurons (i.e. reservoir complexity) would be needed.

Our focus here is on a pattern recognition task based on sequences of sine waves at different frequencies, which best highlights the nonlinearity and transient dynamics of the magnon-scattering reservoir. It also emphasizes the modal multiplexing concept, which, as we state above, is a novel concept in magnetism and perhaps beyond. This is the key result of the paper. Naturally, we appreciate the need to benchmark our approach with other reservoir systems, for example, for tasks such as nonlinear time series prediction and spoken digit recognition. Such studies are planned and require further development but remain beyond the scope of this work.

There are a few ambiguities in the paper that should be more explicitly explained.

Most importantly some results are obtained from simulations others are from experiments. I myself could not always follow what comes from where. Would it be possible to devote a short paragraph to clarify this, and/or carefully state in each figure caption what comes from simulations and what does not?

We carefully checked each figure caption to clarify what data results from experiments and what from simulations. Additionally, we carefully checked

Helmholtz-Zentrum
Dresden-Rossendorf e. V.

Address:
Bautzner Landstr. 400
D-01328 Dresden
<http://www.hzdr.de>

Board of Directors:
Prof. Dr. Sebastian M. Schmidt
Dr. Diana Stiller

Company Registration Number:
VR 1693, Amtsgericht Dresden

Bank Details:
Commerzbank AG
Account No. 0402 657 300
(Bank Code 850 800 00)
BIC DRESDEFF850
IBAN DE42 8508 0000 0402 6573 00

VAT-ID-No.: DE140213784

the manuscript to better highlight in the text when the experiments and the simulations are discussed, respectively.

*The abstract claims 95% accuracy while the text claims 99.4% at the end.
Please clarify what comes from which model.*

In the abstract, we mention recognition rates “above 95%”. We rephrased the sentence to avoid this ambiguity.

Also, reservoir computing devices are notably tricky to characterize: even a very poorly-performing reservoir may seem to achieve high accuracy because of the output linear classifier alone works rather well for many problem classes. I understand that the authors check the role of scattered vs directly accessed nodes to understand this problem – perhaps this argument should be a bit more explicit.

Indeed, we are fully aware of this issue, which is why we made some effort to show how linear classification fails. The first example is shown in Fig. 1 to highlight the fact that the temporal sequences “AB” and “BA” give (nearly) identical spectra for the excited modes but distinct spectra for the scattered modes. This is further explored in simulation, the results of which are shown in Fig. 4. We revised the discussion related to Figure 1 to emphasize this point further. Furthermore, we added the following sentence to the discussion related to Fig 4.

Note that, here, analyzing the directly-excited modes does not correspond to a linear classifier, as these modes themselves experience nonlinear feedback (amplitude losses, frequency shift, etc.) above their power threshold for 3MS.

Overall, I very much enjoyed reading the paper and the scientific quality of the work would deserve Nat. Comm. publication. I have more reservations about the practicality of the device and what future and impact it would have in future computing devices. At a minimum these should be addressed by a few additional arguments in the paper.

We again thank the reviewer for their positive review. We hope the revisions made to the manuscript address these points concerning the practicality of our device.

Helmholtz-Zentrum
Dresden–Rossendorf e. V.

Address:
Bautzner Landstr. 400
D-01328 Dresden
<http://www.hzdr.de>

Board of Directors:
Prof. Dr. Sebastian M. Schmidt
Dr. Diana Stiller

Company Registration Number:
VR 1693, Amtsgericht Dresden

Bank Details:
Commerzbank AG
Account No. 0402 657 300
(Bank Code 850 800 00)
BIC DRESDEFF850
IBAN DE42 8508 0000 0402 6573 00

VAT-ID-No.: DE140213784

Reviewer #2

The Reviewer writes:

This paper studies physical reservoir computing utilizing temporal response of a spin vortex to a time-series input signal, where their unique approach is to use time-averaged power spectra of spin motions for machine-learning computing to extract information encoded in input signals. The former part describes experimental results to demonstrate the basic operation principal of their computing scheme, whereas the latter part describes micromagnetic simulation results and reservoir computing demonstrations.

In the first experimental part, the authors prepared a permalloy disk with an outer ring electrode to induce a time-series rf magnetic field as input. Spin motions at 10 different positions were detected by micro-focused Brillouin light scattering and they were averaged to obtain a power spectrum for output. An input is encoded using a specific combination of the amplitude and frequency of a 20-ns-long sinusoidal signal and two inputs "A" and "B" were prepared. From the end time of an input, a time-series spin motion was recorded for 10 ns to obtain a time-averaged power spectrum. Owing to spontaneous three-magnon scattering and characteristic modes of spin vortex, "A" and "B" have different characteristic spectra having a component at the input frequency as well as two components at lower frequencies. Next, "A" and "B" ("B" and "A") are sequentially combined with an overlap time, to prepare "AB" ("BA") input. Interestingly, an additional component that does not appear in "A" or "B" was clearly obtained. Thus, "AB" and "BA" can be distinguished by power spectra. This is the key to their computing.

In the next part, spin motions in the same permalloy disk were simulated using micromagnetics, with taking into account thermal noise at 300 K and input signal fluctuations. In addition to the experimental part, two inputs "C" and "D" were also prepared, and four-series inputs (ex. ABAB) were explored. A characteristic power spectrum was obtained for each input combination. Using power spectra for all the combinations, reservoir computing was performed to classify series inputs. In consequence, a high classification performance is achieved even under moderate noise levels and input fluctuations.

Their original idea to use time-averaged spectra for reservoir computing has a high impact, their experiments are fundamentally solid, and the high classification accuracy is interesting and remarkable. On the other hand, I think that there are some unclear descriptions, lack of explanations for procedures, and speculative outlooks. Hence, I basically recommend for publication if the manuscript is appropriately/satisfactorily revised.

We thank the reviewer for this positive assessment of our work and hope to include enough information in our manuscript to resolve all unclear descriptions and explanations.

(1) To show the overall of the "reservoir computing" research field, Ref. [1] is just described. I think this reference paper is basically liquid state machine developed from neuroscience and a pulse-based computing model. In addition, the word "reservoir computing" covers analog computing models, such as echo state networks, developed from artificial recurrent neural networks. In my opinion, the authors should show "Jaeger, GMD technical report 148, 2001.". This also comes from the fact that the authors used analog computing, not pulse-based computing.

Helmholtz-Zentrum
Dresden-Rossendorf e. V.

Address:
Bautzner Landstr. 400
D-01328 Dresden
<http://www.hzdr.de>

Board of Directors:
Prof. Dr. Sebastian M. Schmidt
Dr. Diana Stiller

Company Registration Number:
VR 1693, Amtsgericht Dresden

Bank Details:
Commerzbank AG
Account No. 0402 657 300
(Bank Code 850 800 00)
BIC DRESDEFF850
IBAN DE42 8508 0000 0402 6573 00

VAT-ID-No.: DE140213784

Physical reservoir computing was derived from reservoir computing, however, the relation between those two computing frameworks is yet unclear. In my opinion, showing references on physical reservoir computing is very helpful for readers, if possible. For example, G. Tanaka et al., Neural Networks 2019, K. Nakajima, Jpn. J. Appl. Phys. 2020.

If the authors should choose one due to the length limit, I think Jaeger is appropriate according to the computational method in this paper.

We thank the reviewer for pointing this out and included the references in the manuscript.

(2) I understand that cross-stimulated three-magnon scattering (3MS) is the key for successful computing. For example, in Fig. 2E, the frequencies of 3MS for "AB" and "BA" are different from each other. What is the physics behind those phenomena? What are the roles of the first and second inputs? In addition, this type of temporal 3MS has been reported so far? In my opinion, the authors should give comments on these, which can also maximize the originality of this work even the explanation of the physics is qualitative (or speculative). I know that analyses on temporal dynamics are usually difficult, but the knowledge can contribute to understanding of physical reservoir computing.

Cross-stimulated 3-magnon splitting has not been reported so far. We added the following explanatory paragraph to the introduction of our paper, highlighting its originality and providing more context with previous work:

In our previous work [L. Körber et al., Phys. Rev. Lett. **125**, 207203 (2020)], we have shown that individual three-magnon splitting channels, e.g., exciting only f_A , can be stimulated below their threshold power and their temporal evolution significantly modified by additionally exciting one of the secondary modes. In Ref. [L. Körber et al., Phys. Rev. Lett. **125**, 207203 (2020)] this stimulation was achieved by magnons propagating in a waveguide adjacent to the vortex disk. Here, as a logical extension, the role of the stimulating magnon is provided by the secondary modes of another (active) three-magnon channel f_B , a process that we refer to here as cross-stimulated three-magnon splitting. The operation of our MSR strongly relies on the fact that the cross-stimulation between f_B and f_A is not reciprocal due to the involved nonlinear transients. In other words, the effect of the channel f_A on the channel f_B via cross-stimulation differs from the feedback of f_B on f_A .

Time-resolved data on 3-magnon splitting was shown in our previous work [L. Körber et al., Phys. Rev. Lett. **125**, 207203 (2020)].

(3) Throughout the paper, the authors do not describe the computing system in terms of reservoir computing framework. A reservoir computing consists of a preprocessing part, a reservoir part, and a readout part. The input encoding corresponds to the preprocessing part, and the disk corresponds to the reservoir part. These are easy to understand despite of explicit explanations. However, "a power spectrum" corresponds to post signal processing that is contained in the readout part. Also, the worst thing is that the authors do not explain how to obtain the system output class, particularly, there is no description about the readout part. In general sense, there is an output layer with the same size as the target combination numbers, all the reservoir outputs are connected to each system output node, and there are connection weights for all the con-

Helmholtz-Zentrum
Dresden-Rossendorf e. V.

Address:
Bautzner Landstr. 400
D-01328 Dresden
<http://www.hzdr.de>

Board of Directors:
Prof. Dr. Sebastian M. Schmidt
Dr. Diana Stiller

Company Registration Number:
VR 1693, Amtsgericht Dresden

Bank Details:
Commerzbank AG
Account No. 0402 657 300
(Bank Code 850 800 00)
BIC DRESDEFF850
IBAN DE42 8508 0000 0402 6573 00

VAT-ID-No.: DE140213784

nections. In S.M., the authors just describe “we used the classify function”, but readers cannot understand the entire system design. Also, to classify some labels, there are many methods to use, for example, setting one hot vector, the maximum output level in the testing phase is selected, and finally its class label is used for the prediction class. In the present style, readers including me do not understand what are done in the machine learning computing. Hence, I think the authors should describe a simple explanation in the main text and details in S.M..

We apologize if technical details related to the reservoir computing paradigm were not made clearer in the manuscript, but it was not our intention to include such details when the main focus is on the demonstration of a novel concept, i.e., one based on modal-multiplexing in reciprocal space. As such, we tried to highlight the underlying physics behind this concept, since nonlinearities in the magnonics community, for example, are often restricted to fundamental studies (e.g., chaotic dynamics), or avoided altogether (e.g., applications in signal processing or Boolean logic). Our goal, therefore, was to show how through 3-magnon scattering the input sequences, comprising radial modes, can be nonlinearly transformed into an output space, comprising azimuthal modes, on which machine learning can be performed.

We agree that the spin-wave power spectrum, either measured experimentally through Brillouin light scattering or determined numerically in simulation through a discrete Fourier transform of time series data, constitutes the “readout” part of the reservoir. The main point is that the power spectrum is a measure of the respective populations of the scattered modes, which represent the “true” physical output space.

To reiterate our approach: we use frequency binning of the scattered mode spectra to create a discrete output vector (of varying dimensions N) for each input 4-symbol sequence. The different sequences then result in clusters of points in this N -dimensional space, and approaches such as logistic regression are employed to classify these points and develop a trained model. The accuracy of the pattern recognition is then obtained by pitting this model against a test set.

As the Supplementary shows, we used a number of different algorithms for the machine learning part, such as logistic regression, but the actual methods used do not change the overall performance of the reservoir. Admittedly, our rudimentary approach may not be as sophisticated as what the Reviewer invokes, but we believe it is sufficiently robust to highlight the performance of our reservoir. More advanced machine learning approaches will be employed as progress toward device integration is made (e.g., through electrical readout with an ensemble of magnetoresistive sensors), but at this stage, we feel that the current approach conveys the main features of the modal multiplexing idea.

(4) In Fig. 3C, there are many spectra having different shapes. I think it is very helpful for readers to assign the feature induced by 2-series inputs on each typical peak, for example, “AB” and “BA”. This is because the computing seemingly captures the spectrum characterized by the peak amplitude differences and readers must want to simply know what are different.

We tried to follow the suggestion by the reviewer and color-coded the peaks in the nonlinear response which purely result from “A” or “B”, respectively. However, it is difficult to color all peaks since, when cross-stim-

Helmholtz-Zentrum
Dresden–Rossendorf e. V.

Address:
Bautzner Landstr. 400
D-01328 Dresden
<http://www.hzdr.de>

Board of Directors:
Prof. Dr. Sebastian M. Schmidt
Dr. Diana Stiller

Company Registration Number:
VR 1693, Amtsgericht Dresden

Bank Details:
Commerzbank AG
Account No. 0402 657 300
(Bank Code 850 800 00)
BIC DRESDEFF850
IBAN DE42 8508 0000 0402 6573 00

VAT-ID-No.: DE140213784

ulated 3-magnon scattering happens, one of the two secondary modes of “A” will also be populated by scattering from “B”, and vice versa. We modified Fig. 1F to make this more clear. Furthermore, in the 4-symbol patterns, it does not make too much sense to color peaks stemming from “AB” and “BA” since the transient between all of the inputs matters, e.g., “AABA” contains both “AB” and “BA” so that disentangling the syllables is difficult.

(5) In Fig. 4A, the spectrum for “ABAB” is shown. However, I think this is different from the spectrum in Fig. 3C. Please check this. Also, why the frequency range in Fig. 3C is different from that in Fig. 4A? The same range should be appropriate.

In Fig. 3C, we show the nonlinear response only (as noted underneath the x-axis of the figure) and omit the directly excited modes. Moreover, Fig. 3C plots experimental data, and Fig. 4A simulation data, as is mentioned in the figure next to the label “A” and in the caption. Differences in the spectra can be attributed to the different evaluation spaces: In the experiments, we measure at 10 locations only and not the complete disk, as was evaluated in the micromagnetic simulations. Additionally, depending on the excitation of the primary modes, the amplitude distribution of the secondary modes is slightly different. Since it’s difficult to derive the absolute value of the excitation field generated in the experiment, the excitation powers may slightly differ between the simulation and the experiment, leading to subtle differences in the nonlinear response.

We added a sentence in each caption to clarify what is an experiment and a micromagnetic simulation.

(6) In page 5, in the right column, in the first paragraph “Hence, an extension of the input space to more than two frequencies/symbols, or even to more broadband signals, is straightforward.”

This is unclear to me. If the authors claim that they can use 3 or more input expressions for an input sequence (ex. “ABCD”), I do not agree with this. If the authors claim that they can use large numbers of inputs in an input sequence (ex. “ABABABAB”), I do not agree with this. For the former, this is because the computing basically uses temporal nonlinear phenomena and thus one cannot tell what happens when unknown series inputs are used for an input sequence. In the manuscript, the authors just use two inputs for input sequences, which corresponds to binary bit computing. Hence, the authors did not clearly show that their computing method works well for fully analog computing, and thus the outlook is too speculative. For the latter, this is because a power spectrum is calculated using a 10-ns-long transient signal. If there are many sequences, some components will be overlap and/or long-past memories have relatively smaller components. The authors demonstrated only four inputs sequences. Hence, the authors did not clearly show that their computing method works well for long-range input sequences and thus the outlook is too speculative.

I would like the authors to describe more clearly what they want to say.

Fig. R2 below shows the simulation power spectral density of the scattered spin-wave modes for the 24 different combinations of 3-symbol and 4-symbol sequences for a single realization of the thermal noise. Here, each symbol ‘A’, ‘B’, ‘C’, or ‘D’ represents a particular input frequency (and RF field amplitude) with the same pulse durations and overlaps as those

Helmholtz-Zentrum
Dresden-Rossendorf e. V.

Address:
Bautzner Landstr. 400
D-01328 Dresden
<http://www.hzdr.de>

Board of Directors:
Prof. Dr. Sebastian M. Schmidt
Dr. Diana Stiller

Company Registration Number:
VR 1693, Amtsgericht Dresden

Bank Details:
Commerzbank AG
Account No. 0402 657 300
(Bank Code 850 800 00)
BIC DRESDEFF850
IBAN DE42 8508 0000 0402 6573 00

VAT-ID-No.: DE140213784

Fig. R2: Simulated power spectral density (PSD) of the scattered modes for (a) 3-symbol and (b) 4-symbol input sequences, where $f_A = 6.5$ GHz (3.0 mT), $f_B = 7.4$ GHz (3.5 mT), $f_C = 8.9$ GHz (3.0 mT), and $f_D = 10.7$ GHz (3.0 mT), as considered in the main text.

considered in the main text. While far from being an extensive study, one can see by inspection that the different sequences result in *qualitatively different* spectra (although a more thorough quantitative analysis still needs to be done). This validates our remarks on the possibility to extend the processing to more complex sequences. We have included these figures in the Supplementary document [Figs. S7 and S8]. For this we have added the following sentence:

To this end, the supplementary information contains the measured and simulated distinct nonlinear responses for different permutations of the three and four-frequency sequences "ABC" and "ABCD".

Concerning arbitrarily long sequences, such as "ABABABAB", we agree that the present results do not allow us to generalize the approach to arbitrarily long sequences. This issue is the subject of further exploration and will be considered in a future publication.

(7) Throughout the manuscript, the authors do not describe important features for reservoir computing: nonlinearity, short-term memory, and variability. Non-

Helmholtz-Zentrum
Dresden-Rossendorf e. V.

Address:
Bautzner Landstr. 400
D-01328 Dresden
<http://www.hzdr.de>

Board of Directors:
Prof. Dr. Sebastian M. Schmidt
Dr. Diana Stiller

Company Registration Number:
VR 1693, Amtsgericht Dresden

Bank Details:
Commerzbank AG
Account No. 0402 657 300
(Bank Code 850 800 00)
BIC DRESDEFF850
IBAN DE42 8508 0000 0402 6573 00

VAT-ID-No.: DE140213784

linearity do not mean “nonlinear phenomena”. It is defined by nonlinear temporal transformation of input vector to reservoir output vector. This is usually proven by nonlinear tasks, such as temporal XOR. Variability is more difficult to understand for physical reservoir computing researchers, so usually it is not discussed. I think those two are difficult to discuss in the present reservoir computing system and input sequences, and thus it is OK to neglect them. However, I think short-term memory plays an important role in the present system. For example, “ABBA” has a different spectral feature from “AABA”, meaning that first “AB” and “AA” are memorized in their spectra, so I think the memory at least three-step past is contained, i.e., memory length more than 3. Like this, please discuss the results in the context of reservoir computing. (Actually, if one cannot show success in nonlinear tasks, it cannot be called true “reservoir computing”.)

We are puzzled by the Reviewer’s remarks on the nonlinearity, as 3-magnon scattering is precisely the kind of nonlinear transformation that allows us to perform pattern recognition. For example, obtaining different power spectra of the scattered modes for different inputs ‘AB’ and ‘BA’ is the kind of nonlinear transformation (involving transients) that the Reviewer mentions. On this point, we believe we have clearly demonstrated success in a nonlinear task and, therefore, our reference to “reservoir computing” is justified.

Concerning other reservoir computing metrics such as short-term memory (STM), we have not included a discussion on this in the main text as we feel it is a technical point whose usual characterization may not fully apply to our system. The reason for this lies in the readout mechanism: the power spectra are acquired throughout the duration of the input sequences, which is different from trying to reconstruct past sequences based on the output of the present pulse. This is a feature of the model multiplexing scheme.

However, we have performed such tests both in the experiment and in simulation. In both cases, we have used a random, binary AB sequence of 1000 pulses ($f_A = 7.4$ GHz, $f_B = 8.9$ GHz, pulse duration of 14 ns, no overlap) as an input $u(T)$ to the reservoir and acquired the PSD of the scattered modes. The output state of the reservoir, $x(T)$, represents the PSD averaged over the duration of the pulse at time T . The STM task involves predicting inputs from the past based on the present state of the reservoir, where we can define the target output $\hat{y}_{\text{out}}(T; \tau)$ as the input with a time delay τ ,

$$\hat{y}_{\text{out}}(T; \tau) = u(T - \tau). \quad (\text{STM})$$

Training involves constructing the matrix \mathbf{W} such that

$$\mathbf{W} x(T) = y_{\text{out}}(T) \approx \hat{y}_{\text{out}}(T).$$

Here, we have used 800 pulses for training and 200 pulses for testing. In Fig. R3(a)-(c), we show the target and output values for the experimental reservoir for three values of the delay. We can see that reconstruction of the output signal works reasonably well for the first few values of the delay, which indicates that there is short-term memory in the system. This can be quantified with the square of the correlation coefficient,

Helmholtz-Zentrum
Dresden–Rossendorf e. V.

Address:
Bautzner Landstr. 400
D-01328 Dresden
<http://www.hzdr.de>

Board of Directors:
Prof. Dr. Sebastian M. Schmidt
Dr. Diana Stiller

Company Registration Number:
VR 1693, Amtsgericht Dresden

Bank Details:
Commerzbank AG
Account No. 0402 657 300
(Bank Code 850 800 00)
BIC DRESDEFF850
IBAN DE42 8508 0000 0402 6573 00

VAT-ID-No.: DE140213784

Fig. R3: Short-term memory (STM) and parity check (PC) tests of the experimental magnon-scattering reservoir. (a)-(c) Comparison between the target and trained reservoir outputs for three delays τ for the STM task. Square of the correlation coefficient for the (d) STM and (e) PC tasks.

Fig. R4: Short-term memory (STM) and parity check (PC) tests of the simulated magnon-scattering reservoir. (a)-(c) Comparison between the target and trained reservoir outputs for three delays τ for the STM task. Square of the correlation coefficient for the (d) STM and (e) PC tasks.

$$r^2 = \frac{\text{COV}(y_{out}, \hat{y}_{out})^2}{\text{COV}(y_{out}, y_{out}) \times \text{COV}(\hat{y}_{out}, \hat{y}_{out})},$$

where $\text{COV}(x, y)$ is the covariance between two vectors x and y . This value is shown as a function of delay in Fig. R3(d) for the STM task. With the same data, we also performed the same analysis for the parity check task,

$$\hat{y}_{out}(T; \tau) = u(T) \oplus u(T-1) \oplus \dots \oplus u(T-\tau), \quad (\text{PC})$$

where \oplus designates the XOR function. The square of the correlation coefficient for the PC tasks is shown in Fig. R3(e). As these results show, the

Helmholtz-Zentrum
Dresden-Rossendorf e. V.

Address:
Bautzner Landstr. 400
D-01328 Dresden
<http://www.hzdr.de>

Board of Directors:
Prof. Dr. Sebastian M. Schmidt
Dr. Diana Stiller

Company Registration Number:
VR 1693, Amtsgericht Dresden

Bank Details:
Commerzbank AG
Account No. 0402 657 300
(Bank Code 850 800 00)
BIC DRESDEFF850
IBAN DE42 8508 0000 0402 6573 00

VAT-ID-No.: DE140213784

magnon scattering reservoir clearly exhibits short-term memory and is capable of performing nonlinear tasks like XOR.

Fig. R4 shows the same analysis performed for the simulated magnon reservoir, where we can observe similar metrics for the STM and PC tasks. These results further lend support to our characterization of the magnon-scattering reservoir as a reservoir computing. However, we do not feel that they warrant a prominent discussion in the main text, since, as we mentioned above, the particular read-out mechanism of our modal-multiplexing differs from traditional spatial- and temporal multiplexing techniques. To not distract from the main message of this paper, we believe that these benchmarks should be the subject of a forthcoming publication.

**Helmholtz-Zentrum
Dresden–Rossendorf e. V.**

Address:

Bautzner Landstr. 400
D-01328 Dresden
<http://www.hzdr.de>

Board of Directors:

Prof. Dr. Sebastian M. Schmidt
Dr. Diana Stiller

Company Registration Number:

VR 1693, Amtsgericht Dresden

Bank Details:

Commerzbank AG
Account No. 0402 657 300
(Bank Code 850 800 00)
BIC DRESDEFF850
IBAN DE42 8508 0000 0402 6573 00

VAT-ID-No.: DE140213784

Reviewer #3

The Reviewer writes:

L. Korber et.al. reported a pattern recognition method using a magnon reservoir. By using a nonlinear 3-magnon scattering, the authors realized recurrent system in single magnetic disk. The modal multiplexing was clearly detected by using time resolved micro-BLS measurement. The recognition accuracy up to 95% was presented. The wave-based reservoir is important research field of magnonics, and this result is interesting. The contents of this paper are well organized and understandable. The reviewer inquires information to judge its experimental validity. The points are listed below.

We thank the Reviewer's appraisal and interest in this work.

1. For the modal multiplexer, the probe-position dependence of recognition accuracy appears to be of small importance. However, if the probing position was changed, what occurs for the recognition accuracy? If the modal or magnon states was affected by nonlinear 3-magnon scattering, the wave patterns could be different for probing positions and thus the BLS spectrum also could be changed. The presented data were probed by symmetrical positions given in SM file; however, the antisymmetric position appears to increase the accuracy. On the contrary, if the symmetrical position was better, the wave patterns appear to be a steady and less affected by the nonlinear effect (thus history). Does this mean that the system works as physical reservoir with small history?

The Reviewer raises an interesting and important point on the role of the outputs on the accuracy of the pattern recognition. What we have demonstrated in simulations is that the scattered mode spectra provide a suitable nonlinear transformation of the inputs that allows for accurate recognition of AB sequences. In the experiments, we are limited by factors such as the acquisition time, frequency, and spatial resolutions, yet the measured spectra of the scattered modes still exhibit clear differences between the different 4-symbol AB sequences. Given that the output space comprises the azimuthal spin-wave modes, it is expedient to choose a probe position that allows for the largest number of these modes to be detected. Similar considerations will be required when we will attempt electrical readouts, as we discuss in the rebuttal to Reviewer 1 above and further below. Overall, we are confident that as long as the azimuthal modes can be detected experimentally, we should be able to achieve good performance with the magnon reservoir.

Regarding the history, we have recently performed additional characterization of the magnon reservoir using the short-term memory and parity-check tests. These are shown in Figs. R3 and R4 above in this rebuttal. We show that the reservoir possesses short-term memory and is capable of performing nonlinear tasks, with some metrics that are comparable to other reservoir systems. However, we do not feel that these metrics should feature in this paper as the modal-multiplexing scheme differs from the spatial and temporal multiplexing schemes commonly studied. They could, however, be the subject of a forthcoming publication.

2. For the physical reservoir, the output is also important. In Fig. 1F, the output node is not shown. For the actual system, the laser-based output or detection

Helmholtz-Zentrum
Dresden-Rossendorf e. V.

Address:
Bautzner Landstr. 400
D-01328 Dresden
<http://www.hzdr.de>

Board of Directors:
Prof. Dr. Sebastian M. Schmidt
Dr. Diana Stiller

Company Registration Number:
VR 1693, Amtsgericht Dresden

Bank Details:
Commerzbank AG
Account No. 0402 657 300
(Bank Code 850 800 00)
BIC DRESDEFF850
IBAN DE42 8508 0000 0402 6573 00

VAT-ID-No.: DE140213784

method appears to be difficult (BLS or Kerr equipment for single disk). The importance of this paper is the observation of multiplexer in the reciprocal space. However, if the authors insist this method is simple and requires very little pre-processing, the authors discuss the complexity of detection (output) node for modal multiplexing.

The output node comprises the scattered mode spectra, which may be detected in a number of different ways. The use of micro-focus Brillouin light scattering spectroscopy here is an efficient means to probe the spin wave modes directly (“seeing is believing”, as the old adage goes), which is important for a first demonstration of a new concept. Naturally, we expect a future device to rely on an electrical readout, such as with magnetoresistive sensors, but this is far beyond the scope of the present work. As we mention in the rebuttal to Reviewer 1, an electrical read-out scheme is the focus of an EU-funded project and is still a work in progress. Regarding the commercial availability of high-density MRAM storage, which we envision to be modified for electrical read-out, we believe that our magnon-scattering reservoir holds the potential to be practically useful. To comment on this issue we have added the following sentence to our conclusions:

Although our current read-out scheme is based on optical methods, magnetoresistive sensors hold promising possibilities for an all-electric read-out.

3. In Fig. 4B, some description is lacked for f_A , f_C (no frequency information was given).

The frequencies in Fig. 4B are the same as in Fig. 4A. We changed the legend in Fig. 4B to make this clearer.

4. The authors considered the amplitude noise about 3.0mT. Whys this value was picked up? The amplitude dependence of accuracy should be discussed.

The amplitude of 3.0 mT for the RF field excitation (3.5 mT for the excitation at 7.4 GHz) was chosen to ensure that the input drive is above threshold for 3-magnon scattering, which we already confirmed in our previous works [K. Schultheiss et al., *Phys. Rev. Lett.* **122**, 097202 (2019)] and [L. Körber et al., *Phys. Rev. Lett.* **125**, 207203 (2020)].

However, the amplitude of the excitation has not been optimized yet, neither in the experiment nor in the simulation. Moreover, changing the excitation power has a direct impact on the overall nonlinearity of the system including nonlinear frequency shifts and such. If the excitation power is too low, i.e., below the threshold for 3-magnon splitting, everything will be linear and no pattern recognition will be possible. If the excitation power is too high, higher-order magnon scattering processes, like 4-magnon scattering, will set in. Eventually, the system will reach chaos. Therefore, it is difficult to predict how the accuracy changes as a function of the excitation power and requires a more careful analysis of the magnon-scattering reservoir, which we are currently working on.

Nevertheless, our simulation results in Fig. 4 show that the reservoir performance remains robust with fluctuations of up to 10% in this RF field amplitude, which suggests that the capacity for pattern recognition is not restricted to a narrow range of RF field amplitudes.

Helmholtz-Zentrum
Dresden–Rossendorf e. V.

Address:
Bautzner Landstr. 400
D-01328 Dresden
<http://www.hzdr.de>

Board of Directors:
Prof. Dr. Sebastian M. Schmidt
Dr. Diana Stiller

Company Registration Number:
VR 1693, Amtsgericht Dresden

Bank Details:
Commerzbank AG
Account No. 0402 657 300
(Bank Code 850 800 00)
BIC DRESDEFF850
IBAN DE42 8508 0000 0402 6573 00

VAT-ID-No.: DE140213784

We added two remarks pointing out that the excitation powers/strengths have been chosen to be above the respective power threshold for three-magnon splitting.

**Helmholtz-Zentrum
Dresden–Rossendorf e. V.**

Address:

Bautzner Landstr. 400
D-01328 Dresden
<http://www.hzdr.de>

Board of Directors:

Prof. Dr. Sebastian M. Schmidt
Dr. Diana Stiller

Company Registration Number:

VR 1693, Amtsgericht Dresden

Bank Details:

Commerzbank AG
Account No. 0402 657 300
(Bank Code 850 800 00)
BIC DRESDEFF850
IBAN DE42 8508 0000 0402 6573 00

VAT-ID-No.: DE140213784

Reviewers' Comments:

Reviewer #1:

Remarks to the Author:

The authors provided relevant answers to my questions (and as far as I can judge, to other reviewer questions as well). I like the idea of electrically interconnecting a number of magnetic disks to increase the internal complexity of the reservoir - I hope the authors will have the chance of working this idea out in the EU project mentioned in their letter!

The authors carefully discuss the nonlinear scattering mechanisms that enable reservoir computing. There is a vast literature these days on various forms of reservoir computing, but this paper is one of the very few that meaningfully connects the physics to the computing capabilities to the device.

Reviewer #2:

Remarks to the Author:

In my opinion, the authors have precisely revised the manuscript according to the reviewers' comments and provided sufficient explanations. They have also demonstrated additional studies to show new data that can give more deep insight into physical reservoir computing, in answer to my questions. Such results enhance the quality and maximize the originality of their work. Thus, I recommend the manuscript for the publication.

One thing I would the authors like to change is that, in short-term memory and parity check tasks, the square determination coefficient for " $\tau = 0$ " is trivial, so you do not have to plot it in the graphs. Indeed this data point does not have influence on the total capacity, but it is wrong in terms of computing. (sometimes physical RC reserchers plot that data point)

Reviewer #3:

Remarks to the Author:

L. Korber et.al. responded sincerely and discussed the raised points.

Only thing I have still wonder is that the magneto-resistive sensor can be useful for the detection of modal multiplexing.

The presented route is too optimistic (and also I understand it is out of scope of this paper).

Totally the reviewer satisfied the correction and communication, and I recommend the publication of this paper.

Reviewer #1

„The authors provided relevant answers to my questions (and as far as I can judge, to other reviewer questions as well). I like the idea of electrically inter-connecting a number of magnetic disks to increase the internal complexity of the reservoir - I hope the authors will have the chance of working this idea out in the EU project mentioned in their letter!

The authors carefully discuss the nonlinear scattering mechanisms that enable reservoir computing. There is a vast literature these days on various forms of reservoir computing, but this paper is one of the very few that meaningfully connects the physics to the computing capabilities to the device.“

We are happy that we could address all the remaining criticism of our manuscript and thank the reviewer for the kind words.

Reviewer #2

„In my opinion, the authors have precisely revised the manuscript according to the reviewers' comments and provided sufficient explanations. They have also demonstrated additional studies to show new data that can give more deep insight into physical reservoir computing, in answer to my questions. Such results enhance the quality and maximize the originality of their work. Thus, I recommend the manuscript for the publication.

One thing I would the authors like to change is that, in short-term memory and parity check tasks, the square determination coefficient for " $\tau = 0$ " is trivial, so you do not have to plot it in the graphs. Indeed this data point does not have influence on the total capacity, but it is wrong in terms of computing. (sometimes physical RC reserchers plot that data point)“

We thank the reviewer very much for the appraisal of our manuscript and are happy that we could address all comments. Furthermore, we appreciate the recommendation to remove that $\tau = 0$ datapoint from the fading-memory and parity-check plots. However, as mentioned in our previous response, we do not include this figure in the current paper because we believe it would be suited better as part of a forthcoming publication. Therefore, we thank the reviewer again for this valuable remark and will implement it in future work.

Reviewer #3

„L. Korber et.al. responded sincerely and discussed the raised points. Only thing I have still wonder is that the magneto-resistive sensor can be useful for the detection of modal multiplexing. The presented route is too optimistic (and also I understand it is out of scope of this paper). Totally the reviewer satisfied the correction and communication, and I recommend the publication of this paper.“

We appreciate the positive recommendation on our paper and are happy that we could address all comments made. We agree that the proposed magneto-resistive read-out scheme is challenging, yet very exciting.

Helmholtz-Zentrum
Dresden-Rossendorf e. V.

Address:
Bautzner Landstr. 400
D-01328 Dresden
<http://www.hzdr.de>

Board of Directors:
Prof. Dr. Sebastian M. Schmidt
Dr. Diana Stiller

Company Registration Number:
VR 1693, Amtsgericht Dresden

Bank Details:
Commerzbank AG
Account No. 0402 657 300
(Bank Code 850 800 00)
BIC DRESDEFF850
IBAN DE42 8508 0000 0402 6573 00

VAT-ID-No.: DE140213784